# Unsteady State Lightweight Iris Certification Based on Multi-Algorithm Parallel Integration

**Liu Shuai** [1,2] , **Liu Yuanning** [1,2] , **Zhu Xiaodong** [1,2,*] , **Zhang Kuo** [1,2] , **Ding Tong** [2,3] , **Li Xinlong** [2,3] and **Wang Chaoqun** [2,3]

1   College of Computer Science and Technology, Jilin University, Changchun 130012, China
2   Key Laboratory of Symbolic Computation and Knowledge Engineering of Ministry of Education, Jilin University, Changchun 130012, China
3   College of Software, Jilin University, Changchun 130012, China
*   Correspondence: zhuxd@jlu.edu.cn; Tel.: +86-431-8515-9376

**Abstract:** Aimed at the one-to-one certification problem of unsteady state iris at different shooting times, a multi-algorithm parallel integration general model structure is proposed in this paper. The iris in the lightweight constrained state affected by defocusing, deflection, and illumination is taken as the research object, the existing algorithms are combined into the model structure effectively, and a one-to-one certification algorithm for lightweight constrained state unsteady iris was designed based on multi-algorithm integration and maximum trusted decision. In this algorithm, a sufficient number of iris internal feature points from the unstable state texture were extracted as effective iris information through the image processing layer composed of various filtering processing algorithms, thereby eliminating defocused interference. In the feature recognition layer, iris deflection interference was excluded by the improved methods of Gabor and Hamming and Haar and BP for the stable features extracted by the image processing layer, and two certification results were obtained by means of parallel recognition. The correct number of certifications for an algorithm under a certain lighting condition were counted. The method with the most correct number was set as the maximum trusted method under this lighting condition, and the results of the maximum trusted method were taken as the final decision, thereby eliminating the effect of illumination. Experiments using the JLU and CASIA iris libraries under the prerequisites in this paper show that the correct recognition rate of the algorithm can reach a high level of 98% or more, indicating that the algorithm can effectively improve the accuracy of the one-to-one certification of lightweight constrained state unsteady iris. Compared with the latest architecture algorithms, such as CNN and deep learning, the proposed algorithm is more suitable for the prerequisites presented in this paper, which has good environmental inclusiveness and can better improve existing traditional algorithms' effectiveness through the design of a parallel integration model structure.

**Keywords:** lightweight constrained iris; one-to-one certification; unsteady state texture; multi-algorithm parallel integration model; maximum trusted method

## 1. Introduction

Iris recognition is one of the most recognized technologies in the field [1]. In the current application scenario, there is a case where the same category comparison of a test feature is compared with certification information previously stored in the chip card under the lightweight iris scale; that is, in the case of a small-scale iris sample set (the number of irises is small), the test features are stored in the chip card in advance. The feature information is compared to achieve the same category of certification. This kind of scenario requires the algorithm to meet two requirements. First, the accuracy

of single-person recognition is high. The algorithm can distinguish the features among the same category and different categories effectively. The other is to speed up and reduce the certification time as much as possible, as well as the model adjustment time after adding or reducing the category.

Collection person shooting states are unsteady and unpredictable at different times, and the difference of the texture state among the irises of the same person may be too large. In terms of feature extraction and recognition, the traditional iris recognition algorithms stabilize the collection state of the test iris and template iris by controlling the changes of external environment and acquisition state [2], making the iris texture state as consistent as possible, which is suitable for high-end acquisition equipment (high-pixel camera, good signal sensor performance, and auxiliary camera algorithms for long-distance shooting) such as a near-infrared (NIR) camera [3]. For low-end acquisition devices [4] (with low numbers of pixels of only tens of thousands to a million or so, only close-range camera function, general signal sensor performance), this method will increase the difficulty of collecting the iris and make iris quality dependent on the proficiency of the staff using the iris equipment.

In addition, the acquired image can be processed by an algorithm to accurately locate the iris position, extract the iris effective information, and suppress the interference caused by the change of the texture state, including, in addition, the possibility of processing the acquired image using some algorithms to suppress the interference caused by the changes of texture state. The main research direction is heterogeneous iris recognition [5], including fuzzy image correction [6], effective feature region rotation adjustment [7], unconstrained iris localization [8], iris region selection of interest [9], and effective information extraction based on seeded model [10]. This method has high operability and low requirements on the acquisition equipment. It can extract effective features for some multi-state irises, and improve the accuracy of iris certification. However, the algorithm has low flexibility and certain pertinence. Special features need to be combined with special feature extraction and recognition algorithms.

In recent years, the method of multi-algorithm combination has also been a commonly used method. By combining a variety of different feature extraction algorithms and recognition algorithms, the feature differentiation degree of different categories under different acquisition conditions is improved as much as possible, thereby offsetting the iris state change factors. A common situation is the secondary recognition mode [11], which performs recognition through layer-by-layer screening of sequential structures. Although such algorithms can further expand the recognizable range, the quality requirements for images themselves are relatively high. The multi-algorithm parallel mode [12] combines multiple algorithms in parallel and performs decision analysis on different results to obtain the final result. Compared with one single algorithm, the integrated model can authenticate the iris through various algorithms and use the complementarity and diversity between different certification algorithms to improve the accuracy and robustness of iris certification. These algorithms are highly flexible and can be freely matched. There are a variety of different feature extraction and recognition algorithms, but the question of how to design the decision-making method is the key, which needs to improve the complementary features between various algorithms in order to be achieved.

In addition, in order to ensure the accuracy of certification, the certification model needs to be trained. The traditional algorithm is statistical training through a large amount of data [13], or small-scale training through label setting [14]. Although the accuracy of the authentication model can be effectively improved, the two methods require a large amount of data to be supported, and the iris data of each case is also distinguished before certification. The data requirements are high, and the training process is complicated.

In summary, in the face of the case that the changes in the shape of the iris texture are caused by changes in iris imaging conditions at different times, currently there are a large number of image processing algorithms, iris feature extraction algorithms, and iris recognition algorithms, but few studies have explored the interrelationship among multiple algorithms. For specific algorithm prerequisites, the existing algorithms are combined and effectively connected, which can greatly improve the accuracy of the algorithm under these prerequisites.

In this regard, in this paper, a multi-algorithm parallel integration general model structure is proposed for the situation of iris shooting conditions changing at different moments, causing iris texture change greatly. Through a three-module structure comprising an image processing layer, a feature recognition layer, and a decision layer, different algorithms are parallel-filled into the integration model based on different prerequisites. The prerequisites of the certification algorithm can be satisfied by a parallel combination of multiple algorithms; thus, the same structure can be used to accurately apply iris certification to different situations through a combination of different algorithms. In addition, the lightweight constrained iris is taken as the research object (iris acquisition instrument remains unchanged and the shooting state standard). Aiming at the impact of collection person's own acquisition status (defocus, deflection) and external environment (illumination) on the iris texture features, some specific algorithms are given to this model structure and a one-to-one certification algorithm for constrained unsteady-state iris based on multi-algorithm integration and maximum trusted decision is designed on the basis of this model.

The image processing layer has six image processing algorithms: Gaussian filter [15], Square filter [16], Median filter [17], Bilateral filter [18], Equalization histogram [19], Laplacian of Gaussian operator [20]. The differences among irises are suppressed from different angles by these six algorithms, stable feature points inside the iris texture are extracted as the iris recognition area, which can reduce the interference of defocus image on the feature. In the feature recognition layer, in order to ensure the certification speed, the feature recognition layer is only composed of two improved traditional methods: first, the Gabor filter group [21] and the Hamming distance [22] (Gabor + Hamming), and second, the Haar wavelet [23] and the BP neural network [24] (Haar + BP). The second method, in addition to improving the speed of certification, has a certain rotation invariance [25], which can eliminate the interference caused by deflection. Because external lighting conditions will affect the algorithm results, it is necessary to perform a reliable analysis of recognition results based on the test statistics under specific external environmental conditions (illumination). Two certification results are input into the decision layer, and the maximum trusted result under experimental illumination conditions is selected as the final decision result, which can suppress the interference of illumination factors on the expression of feature and improve the recognition accuracy of the model.

Compared with other traditional single-certification algorithms, the innovation and design concept of the certification model proposed in this paper are as follows:

1. A multi-algorithm parallel integration general model structure is proposed for the situation that can effectively connect different algorithms, so as to assign specific algorithms to the multiple algorithms parallel structure for specific application scenarios, so that the same structure can be flexibly replaced for different scenarios, and pay attention to the connection problem among different algorithms, which can greatly improve the efficiency of algorithms, so that simple algorithms can also play a powerful role under certain conditions.

2. One-to-one certification is performed by using the lightweight constrained iris with unsteady texture change caused by illumination, defocused, and deflection. Effective and stable iris areas are selected, and two recognition algorithms are used to identify their categories in parallel. Recognition algorithm credibility in certain illumination environment can be obtained by statistics learning, and the conclusion can be obtained based on the credibility of the result of this illumination environment. This algorithm considers collection person status and the impact of external environment change on the iris recognition, so the same algorithm structure can perform iris one-to-one certification to the maximum extent, which has highly environmental inclusion.

Overall, this paper effectively combines the existing algorithms for lightweight iris certification, which can enable multiple algorithms to be better connected in the prerequisites of this paper. While ensuring the accuracy of authentication, the difficulty of training is reduced.

## 2. Multi-Algorithm Parallel Integration Model and Algorithm Prerequisites

The iris image needs to be processed before extracting features. Figure 1 shows the resulting iris image presented after the various steps of image processing.

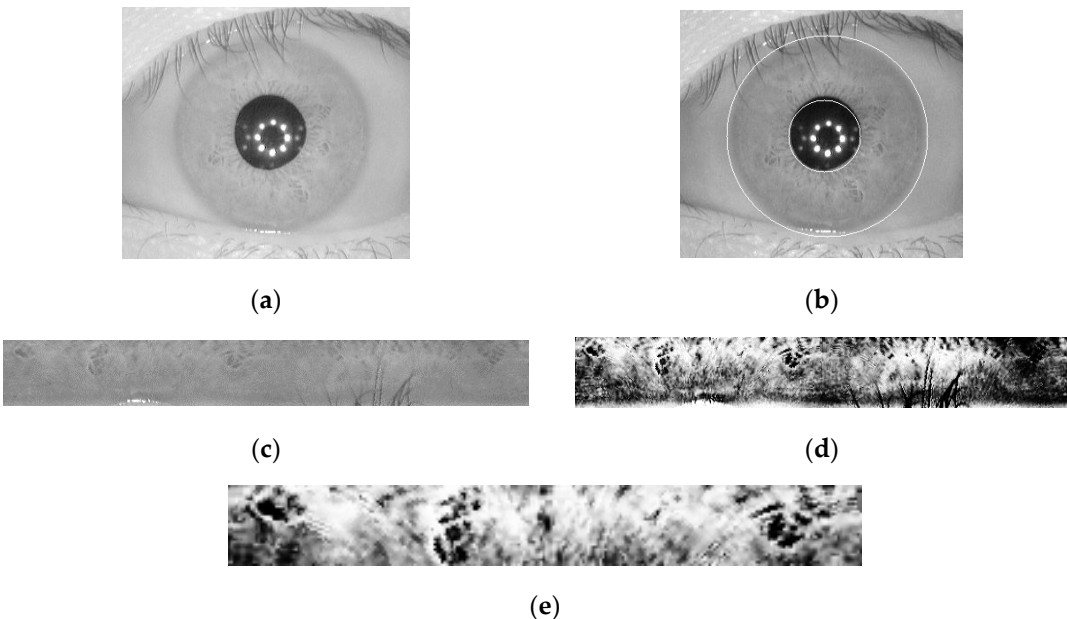

**Figure 1.** Results of each step of the iris-processing stage: (**a**) quality qualified image; (**b**) iris area (two inner regions of the white ring); (**c**) normalized image; (**d**) enhanced image; (**e**) recognition area.

The iris processing steps are as follows:

1. Irises that cannot complete the recognition process are eliminated by quality evaluation [26], which can obtain the quality judgment qualified image (Figure 1a);

2. The iris region with acceptable quality is then separated from the eye image by the positioning algorithm [27], which can determine the position of the iris area (the white ring in Figure 1b);

3. The iris regions of different dimensions are mapped to the normal image of the fixed dimension by the rubber band mapping method [28] (Figure 1c). Finally, the iris textures are highlighted by equalizing the histogram [29] (Figure 1d). In this paper, the normalized iris image is $512 \times 64$ dimensions, and $256 \times 32$ dimensions are taken from the upper left corner as the recognition area (Figure 1e).

In this paper, iris certification is carried out under the extreme situation can be excluded (the shooting content is a living eye) and the inner and outer circle of the iris are correctly segmented.

### 2.1. Multi-Algorithm Parallel Integration Model Structure

The multi-algorithm parallel integration model structure used in this paper is a basic structure, being divided into three layers, including: image processing layer, feature recognition layer, and decision layer. The multi-algorithm parallel integration model structure is shown in Figure 2.

The three layers are independent of each other and affect each other. The image processing layer processes the iris image according to the feature extraction and recognition algorithm of the feature recognition layer, and the appropriate image processing algorithms are used to highlight the iris features that need to be expressed by the feature extraction algorithm, thereby improving the accuracy rate of the iris recognition algorithm. The feature recognition layer expresses the prominent features in different ways, so that each method forms a unique expression pattern for each category of iris, and one-to-one certification is performed by the recognition algorithm corresponding to a certain feature, and then different decision recognition sets are formed. The decision layer selects the most

probable result in the recognition result set as the final output result according to different iris collection conditions and other appropriate decision algorithms.

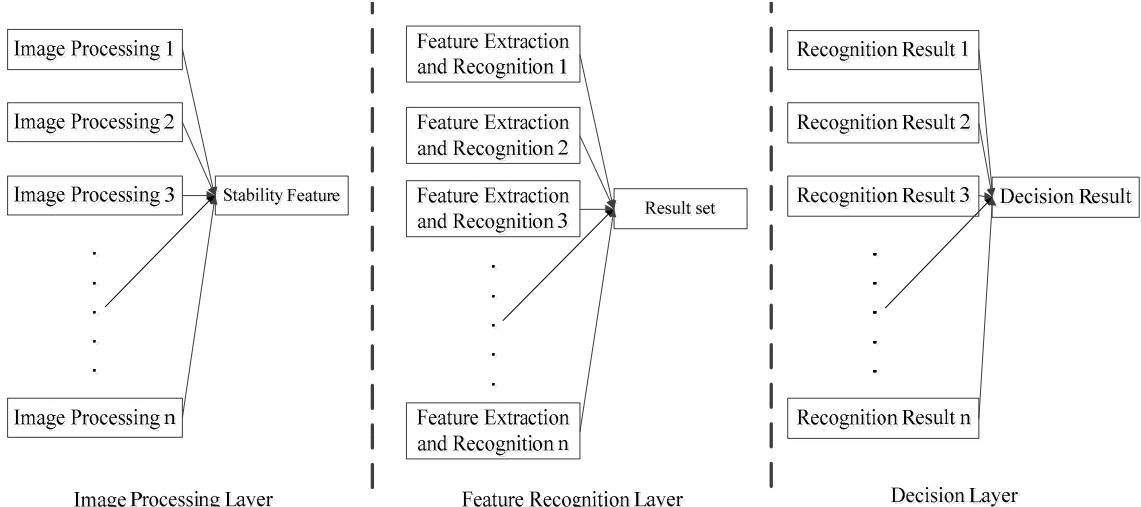

**Figure 2.** Multi-algorithm parallel integration model structure.

Any algorithm has prerequisites for its use, and the algorithm prerequisites for various iris certification algorithms are different. Depending on the specific algorithm prerequisites, different algorithms are populated in each layer to maximize the efficiency of the parallel structure under these prerequisites.

## 2.2. Algorithm Structure and Overall Idea Introduction

There are two main things that affect the recognition effect of the iris recognition algorithm: the algorithm itself and iris quality. The interference factors for iris quality include the external environment of iris collection and the state of the collection person themselves. The external environment includes illumination, equipment parameters, whereas the state of the collection person includes the degree of head deflection, defocus, blinking, and constrained acquisition. Because the collection state at different times cannot be exactly the same, even in the same person the iris states at different shooting time experience a certain degree of change. Examples of eye images of the same person at different shooting times is shown in Figure 3.

An example of the acquisition state change of the collection person being small and the external environment being stable is shown in Figure 4. The images with a steady change in constrained iris texture are taken of the same person.

An example of the state of the collector changing (defocus, deflection) and the external environment being unsteady (illumination) is shown in Figure 5. The images with an unsteady change in constrained iris texture are taken of the same person.

It can be seen from Figures 4 and 5 that, in the case of the same person's state being stable in the external environment, the iris texture image captured at different times is stable and the change is small; where conditions are opposite, the iris texture experiences a large change in state. The main influencing factors are defocus, deflection of human eyes, and external illumination.

Therefore, the algorithm prerequisites set in this article are:

1. Constrained iris (the iris of the recording instrument is unchanged and the shooting state is standard)

2. The collection state of the collection person may change at different times (the appearance of the iris texture is affected by the defocus and positional offset)

3. The external environment may change at different times (the presentation of the iris texture is affected by the intensity of the illumination)

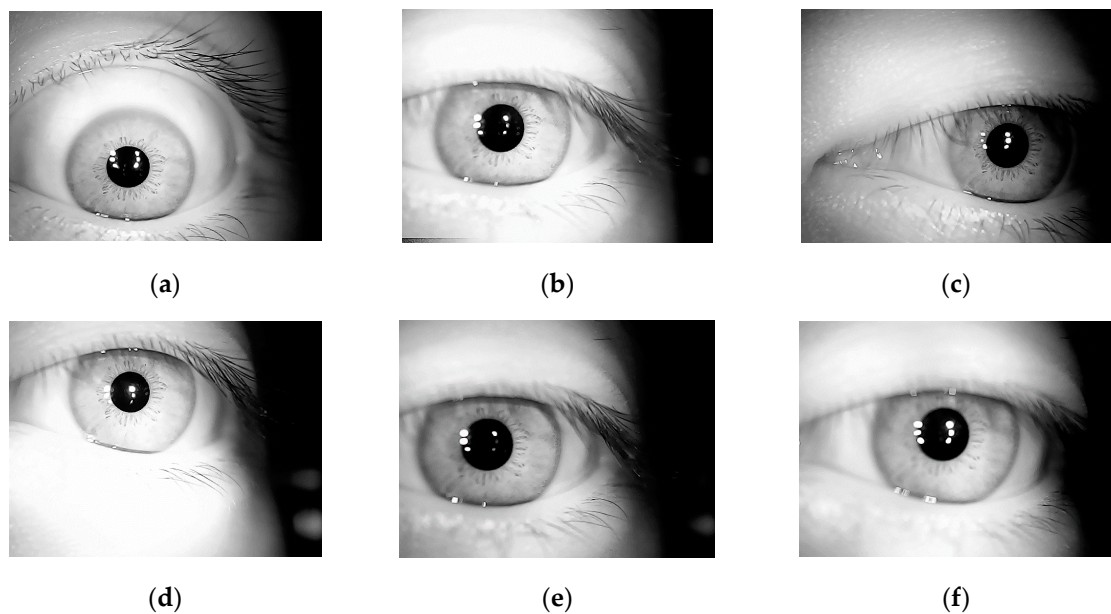

**Figure 3.** Examples of eye images of the same person at different shooting times: (**a**,**b**) clear grayscale image with normal light conditions; (**c**,**d**) clear grayscale image with darker and brighter light conditions, respectively; (**e**,**f**) blurred grayscale images.

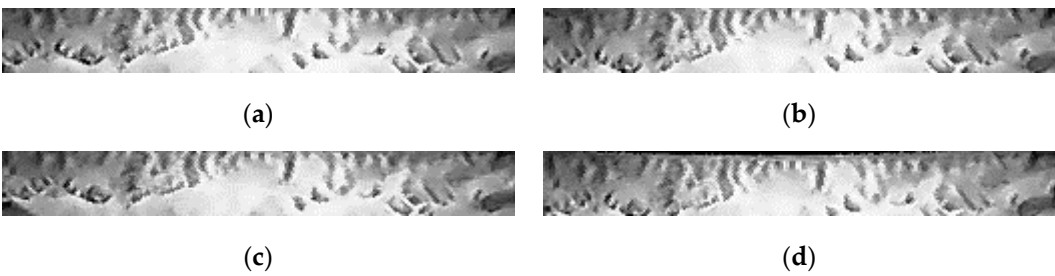

**Figure 4.** Steady change of constrained iris texture taken of the same person.

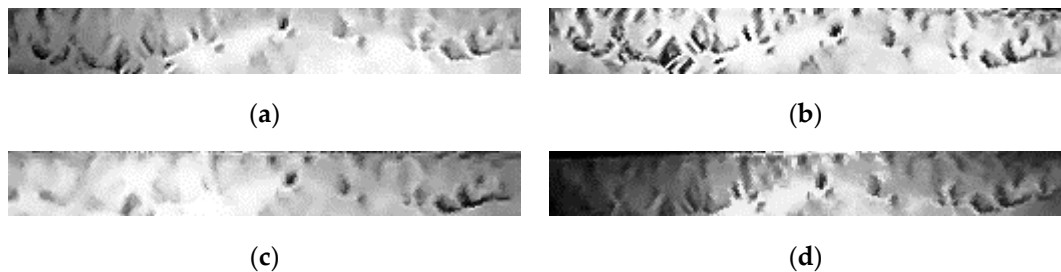

**Figure 5.** Unsteady change of constrained iris texture taken by the same person: (**a**) normal; (**b**) normal illumination clear and undeflected; (**c**) normal illumination blur and deflected; (**d**) dark illumination clear and undeflected.

## 3. Lightweight Unsteady State Constrained Iris Certification

The overall process of the algorithm is shown in Figure 6.

**Image processing layer:** Biometric information focuses on the uniqueness and stability of features, such as the shape and size of facial features in face recognition [30]. The variation of the amount of information in each part of the iris can be used as a unique iris feature [31], because the relative change relationship among the iris key points rarely changes greatly, regardless of the change of the external environment and the acquisition status. Therefore, the stability of the features means that the information measured or obtained is robust to certain distortions.

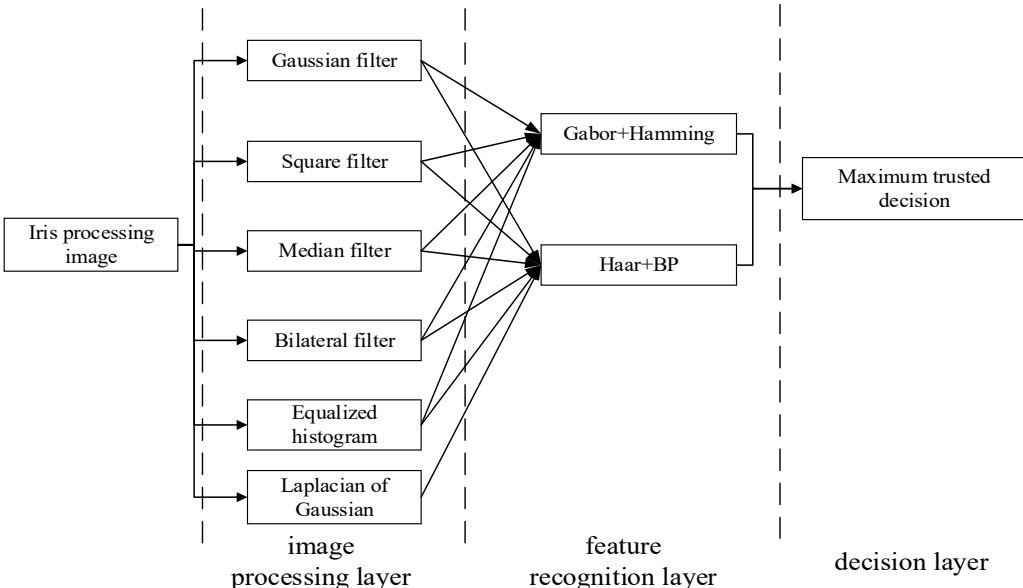

**Figure 6.** Overall process of the algorithm.

Based on these two points, in this algorithm, the stable texture after the iris is subjected to multiple filtering processing is set as an effective feature, which is specifically defined as: feature points existing on each of the filtered processed images in the multi-filter processing. Only the image inside the iris texture can easily be preserved in the image after the edge is highlighted or the image smoothing is processed, from which it is possible to effectively retain enough of the internal feature points of the iris texture reflecting the amount of iris information.

There are two reasons for filtering images: on the one hand, the image is smoothed and iris noise is removed, thereby reducing unnecessary interference; on the other hand, textures will be highlighted, and iris edge information can be saved. To ensure that the points in the iris texture can be fully expressed, in this paper, six filters that can achieve these two purposes are selected in the image processing layer, including: Gaussian filter, Square filter, Median filter, Bilateral filter, Equalization histogram and Laplacian of Gaussian.

Gaussian filter: A linear smoothing filter that performs a weighted average. The value of each pixel is obtained by weighted averaging of its own and other pixel values in the neighborhood.

Square filter: A linear filtering technique where each output pixel is the average of pixel values in the kernel neighborhood.

Median filter: A non-linear smoothing technique that sets the grayscale value of each pixel to the median value of all pixels within a neighborhood window at that point.

Bilateral filter: A non-linear filtering method that combines the spatial proximity of image with the similarity of pixel values. It also considers the spatial information and the gray similarity to achieve the purpose of edge preservation and denoising.

Equalization histogram: The nonlinear stretch is performed on the image, the image pixel values can be reassigned and the number of pixels in a certain grayscale range is approximately the same, which can increase the grayscale dynamic range of image.

Laplacian of Gaussian: optimal smoothing is performed on the original image which can suppress noise.

These six kinds of filters are combined to ensure that the feature points inside the iris texture are preserved, so that the information variation law can still be effectively expressed, and the unstable noise points are removed to ensure the validity of the stable features. The features are then made to conform to the definitions in this paper.

**Feature recognition layer:** Gabor and Haar are the most commonly used iris feature extraction filters. Their performance has been fully tested, and they have certain iris rotation invariance. Because

the method of extracting the information variation law is used in this paper, the Gabor filter and Haar wavelet are more suitable for the image processed by the image processing layer. The Gabor filter performs iris certification by converting the filter direction and frequency into a binary form, which is a carrier code and is more suitable for the Hamming distance model. The Haar wavelet uses the amount of the block information as the iris feature, and the neural network is more suitable for this. The appropriate feature extraction and recognition algorithms are selected for the problem of phase deflection during iris acquisition in the feature recognition layer, and the differences between various categories of irises are improved, the features of the same iris category can be gathered, and the interference of phase deflection can also be suppressed as much as possible.

**Decision layer:** The illumination condition can affect the appearance of iris texture, so that the iris texture change law is not obvious. Although the image processing layer and the feature recognition layer can eliminate some external environmental influences, the algorithm itself is different for the external illumination environment, and the decision layer is different. The function is to set the algorithm to have the maximum number of correct certifications to be the most trusted algorithm in the illumination environment by statistically evaluating the algorithm's recognition results under certain circumstances in a certain external environments (illumination). The certification result of the algorithm is taken as the final certification result under the illumination condition.

### 3.1. Image Processing Layer

The extraction process of effective information (stable features) in the image processing layer is as follows:

1. The iris enhancement images of the same person are separately filtered to obtain six different filtered processed images, as shown in Figure 7.

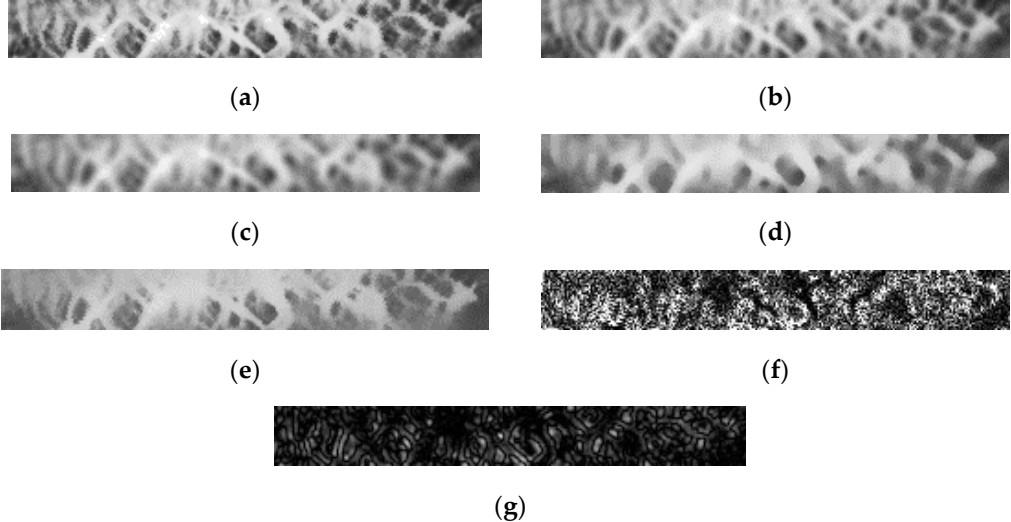

**Figure 7.** Original image and six different filtered processed images: (**a**) Iris original image; (**b**) Gaussian filter; (**c**) Square filter; (**d**) Median filter; (**e**) Bilateral filter; (**f**) Equalization histogram; (**g**) Laplacian of Gaussian.

2. The six filtered processed images are subtracted from the original image respectively to obtain six feature difference images, respectively, as shown in Figure 8.

All points where the gray values are bigger than 0 in the six feature difference images are taken as feature points, and the feature points' gray values of the original image are mapped onto a $256 \times 32$-dimensional image. The gray value of the non-feature points is set to 0. Finally, a stable feature recognition area can be obtained.

The example of stable feature recognition areas between a group of the same category and different category is shown in Figure 9.

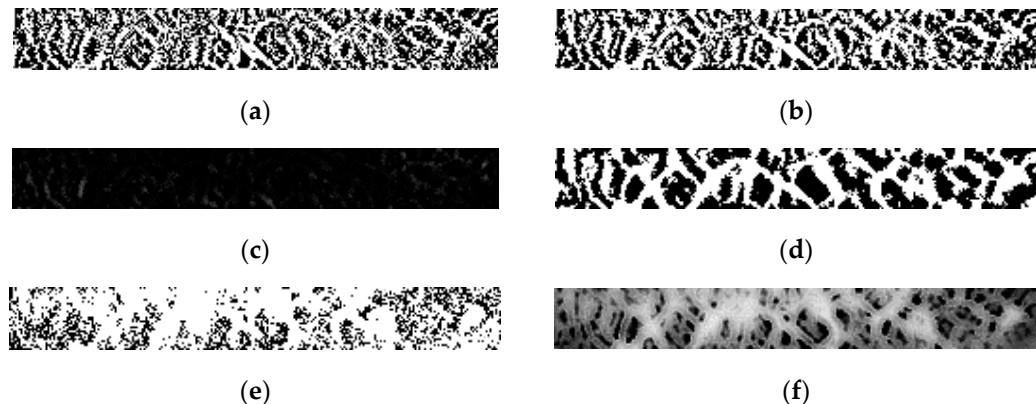

**Figure 8.** Six feature difference images: (**a**) Gaussian filter; (**b**) Square filter; (**c**) Median filter; (**d**) Bilateral filter; (**e**) Equalization histogram; (**f**) Laplacian of Gaussian.

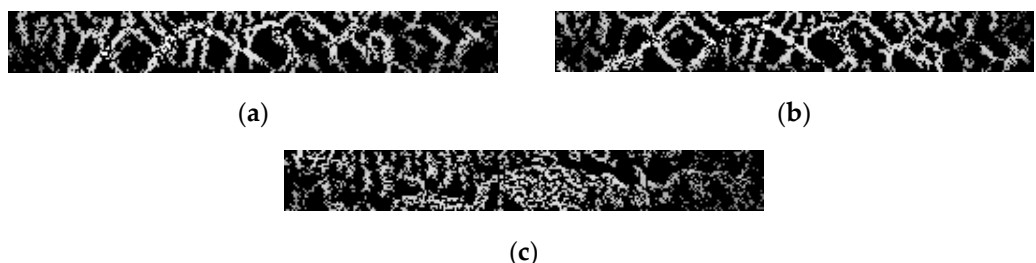

**Figure 9.** Stable feature recognition area of the same category and different category: (**a**) same category 1; (**b**) same category 2; (**c**) different category.

### 3.2. Feature Recognition Layer

Among the iris features, the most common method is the change law of iris texture change. This paper takes the change rule of the amount of iris information as the iris feature. It can be seen from Figure 9 that the texture feature trend expressed in the image processing layer can distinguish between different iris categories effectively; however, there are still some differences in the iris texture of the same category due to the phase change. Therefore, the main goal of feature recognition is to minimize the influence of iris deflection while performing recognition by algorithms with rotation invariance. Two methods are used in this paper: The Gabor filters group and the Hamming distance (Gabor + Hamming); and the Haar wavelet and the BP neural network (Haar + BP). Some improvements based on traditional algorithms are made for better certification according to this paper's prerequisites.

#### 3.2.1. Gabor + Hamming

The Gabor filter expression is shown in Equation (1).

$$\phi(x,y) = \frac{f_0^2}{\pi\gamma\eta} e^{\left(-\left(\frac{f_0^2}{\gamma^2}x_t^2 + \frac{f_0^2}{\eta^2}y_r^2\right)\right)} e^{(-j2\pi f_0 x_r)}, \tag{1}$$

where $x_r = x\cos\theta + y\sin\theta$, $y_r = -x\sin\theta + y\cos\theta$, $f_0$ is the center oscillation frequency; $\theta$ is the rotation angle of the Gabor kernel function; $\gamma = f_0/\alpha$; $\alpha$ is the width of the Gaussian function in the $x$ direction; $x_r$ is the width of the $x$ direction vector at the angle $\theta$ of rotation; $y_r$ is the width under the $y$ direction vector at the angle $\theta$ of rotation; $\eta = f_0/\beta$, and $\beta$ is the width of the Gaussian function in the $y$ direction.

If $y = \eta = \delta/\sqrt{2\pi}$, $\vec{k} = 2\pi f_0 e^{(j\theta)}$, and $\vec{z} = (x,y)$, Equation (1) can be expressed as Equation (2).

$$\phi(\vec{z}) = \frac{1}{2\pi} \frac{\|\vec{k}\|^2}{\sigma^2} e^{\left(-\frac{\|\vec{k}\|^2\|\vec{z}\|^2}{2\sigma^2}\right)} e^{(j\vec{k}\times\vec{z})}, \tag{2}$$

where $\sigma$ is the standard deviation of the Gaussian function; in this paper $\sigma = 2\pi$.

To obtain the variation direction $M$ of the Gabor filter kernel function and the oscillation amplitude $\phi_{m,n}$ on the frequency scale $N$ variable, Equation (2) can be shown as in Equation (3).

$$\phi_{m,n}(\vec{z}) = \frac{1}{2\pi} \frac{\|\vec{k_{m,n}}\|}{\sigma^2} e^{\left(-\frac{\|\vec{k_{m,n}}\|^2 \|\vec{z}\|^2}{2\sigma^2}\right)} e^{(j\vec{k_{m,n}} \times \vec{z})} \vec{k_{m,n}} = k_n e^{(i\varphi m)}, k_n = \frac{k_{\max}}{f^v}, \varphi_m = \pi m/8, \quad (3)$$

where $k_n$ represents the frequency of the Gabor filter; $k_{\max}$ is the maximum frequency of the Gabor filter; $f^v$ represents the frequency difference between two adjacent Gabor cores, $v = 1, 2, \ldots, m$. Finally, there are a total of $m \times n$ Gabor filters for iris feature extraction, where the frequency scale is $m$. The direction $\phi$ of the Gabor filter is equally divided into $n$ parts in the interval [0°,180°]. It can be seen from Equation (3) that the adjustment of the Gabor filter is determined by four parameters $\{k_{\max}, f^v, m, n\}$. In this paper, $k_{\max} = 0.5\pi, f^v = \sqrt{2}$ (value from [32]), $m = 4$, and $n = 4$, giving a total of 16 groups of filters.

The $256 \times 32$-dimension iris recognition region is processed with 16 sets of Gabor filter and the processed image numbers are $G_1$ to $G_{16}$. Each Gabor processed image $G_n$ is equally divided into $8 \times 4$ sub-graphs of $32 \times 8$ dimensions. The sub-graphs are numbered $B_{n-1} \sim B_{n-32}$. The feature code is set by using the over-threshold judgment method. The determination threshold is set $M_1$. The average amplitude value $T$ of each sub-graph is compared with $M_1$ (the decision threshold is calculated from the training iris library that meets the prerequisites for this paper, which is the best value for most iris certifications). If the amplitude value is smaller than or equal to $M_1$, the feature code of this sub-graph is set to 0. If the amplitude value is bigger than or equal to $M_1$, the feature code of this sub-graph is set to 1. Finally, there is a 512-bit ($32 \times 16$) binary feature code. The Hamming distance between the test iris and the template iris is calculated as a feature similarity. The feature similarity calculation formula of Gabor + Hamming is shown in Equation (4).

$$HD = \frac{1}{512} \times \sum_{i=1}^{512} \frac{1 + \text{sgn}(AT_{n-i} - M_1)}{2} \oplus \frac{1 + \text{sgn}(BT_{n-i} - M_1)}{2} \quad n = 1, 2, \ldots, 16 \quad i = 1, 2, \ldots, 32 , \quad (4)$$

where $AT_{n-i}$ denotes the average amplitude value of the $i$-th sub-graph of the $n$-th Gabor processed image in the test iris. $BT_{n-i}$ denotes the average amplitude value of the $i$-th sub-graph of the $n$-th Gabor processed image in the template iris; and $HD$ is the feature value of Gabor + Hamming distance recognition. The Gabor + Hamming distance feature value is compared with the set classification threshold. If it is less than the classification threshold, the test iris and the template iris are considered as the same category.

The function of this recognition algorithm is to binary code the vibration amplitude value of feature texture in the form of sub-graphs by different filters and authenticate the same category through a certain range of iris texture change rules, thereby avoiding the influence of corresponding point feature difference caused by phase difference.

### 3.2.2. Haar + BP

The high-frequency coefficients of the first-layer low-frequency sub-blocks after Haar wavelet processing are extracted from the three sub-blocks G1, G2, G3. These are the wavelet coefficient matrix of the first layer low-frequency sub-blocks after image decomposition. G1 is the horizontal high frequency sub-block, G2 is the vertical high frequency sub-block, and G3 is the diagonal high-frequency sub-block. The dimensions of the three sub-blocks G1, G2 and G3 are $128 \times 16$. The Haar wavelet belongs to the point feature extraction algorithm and the sub-block dimensions are too large. To reduce the dimensions, G1, G2 and G3 are first equally divided into 12 sub-blocks with $64 \times 8$ dimensions. The principal component analysis method [33] is used to reduce the dimension of 12 sub-blocks to $64 \times 1$ dimensions, which is 64 feature points. The average value of 64 feature points is calculated as

the feature value of each sub-block. This meant a total of 12 feature values. The Haar wavelet and the block of each sub-block are shown in Figure 10.

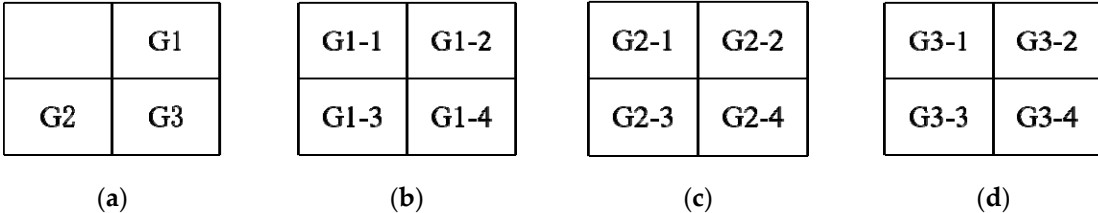

|       | G1   |
|-------|------|
| G2    | G3   |

(**a**)

| G1-1 | G1-2 |
|------|------|
| G1-3 | G1-4 |

(**b**)

| G2-1 | G2-2 |
|------|------|
| G2-3 | G2-4 |

(**c**)

| G3-1 | G3-2 |
|------|------|
| G3-3 | G3-4 |

(**d**)

**Figure 10.** Haar wavelet and the block of each sub-blocks: (**a**) Haar wavelet; (**b**) Sub-block distribution of G1; (**c**) Sub-block distribution of G2; (**d**) Sub-block distribution of G3.

According to the classification of the data, the neural network is designed to use a three-layer neural network, and the number of nodes for G1, G2, and G3 is 12, 4, and 1, respectively. The excitation function is a sigmoid function [34]. The value of connection weights is either 1 or 0, which only serves to transfer data. In Figure 12, the solid line represents the connection weight as 1, and the dotted line represents the connection weight as 0. The input layer is the absolute value of the difference of iris features between the test iris and the template iris, the sub-blocks of the same sequence number in G1, G2, and G3 are added, and the values are input into the hidden layer. The neural network structure is shown in Figure 11.

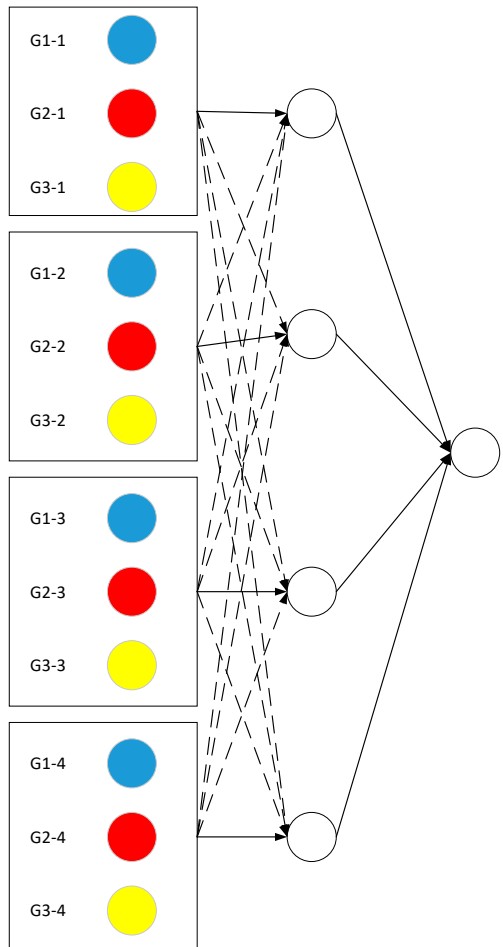

**Figure 11.** Neural network structure.

The hidden layer input value is shown in Equation (5).

$$X_i = |CG1 - i| + |CG2 - i| + |CG3 - i| \quad i = 1, 2, 3, 4 \ , \tag{5}$$

where $X_i$ is the input value of the $i$-th hidden layer node. $CG1_i$ is the difference between $G1 - i$ of the test iris and $G1 - i$ of the template iris; $CG2_i$ is the difference between $G2 - i$ of the test iris and $G2 - i$ of the template iris. $CG3_i$ is the difference between $G3 - i$ of the test iris and $G3 - i$ of the template iris. The input layer output values are input to the hidden layer. The hidden layer output value $S_i$ is shown in Equation (6).

$$\begin{cases} S_i = \frac{1}{1+e^{-D_1 \times X_i}} & X_i > M \\ S_i = \frac{1}{1+e^{-D_2 \times X_i}} & X_i \leq M \end{cases}. \tag{6}$$

The stable texture features of the same category of iris are similar; therefore, the value of each iris feature difference will be small. According to the nature of the Sigmoid function, the smaller the input, the greater the output. For this reason, in order to expand the difference between irises of the same category and irises of different categories, threshold weights are set for the input of the hidden layer. The input value of the hidden layer is compared with the threshold $M$. If the value of $X_i$ is greater than $M$ is multiplied by the threshold weight $D_1(D_1>1)$, making the value of $-D_1 \times X_i$ closer to negative infinity, the closer the hidden layer output value $S_i$ of the $i$-th hidden layer node is to 1; if the value $X_i$ is less than or equal to $M$ multiplied by the threshold weight $D_2(D_2<1)$, making the value of $-D_2 \times X_i$ closer to positive infinity, the closer the hidden layer output value is to 0. In turn, the distinguishing of the hidden layer output can be improved.

Finally, the hidden layer output value is input to the output layer. The input value Z of the output layer is shown in Equation (7).

$$Z = \sum_{i=1}^{4} S_i - \frac{1 - Sgn(S_i - F)}{4} - 4. \tag{7}$$

Because the input values of the hidden layer are all positive, the output values $S_i$ of the hidden layer are all greater than 0.5, and the output values of the different iris categories of the hidden layer are smaller. Therefore, in order to increase the difference between irises of the same category and irises of different categories, threshold $F$ is set. The output value of the hidden layer is compared with the threshold $F$; if the value of $S_i$ is less than or equal to $F$, the output value $S_i$ needs to have 0.5 subtracted from it, which can make a small value become smaller ($Sgn(S_i - F)$ is a symbolic function, if $S_i \leq F$, $Sgn(S_i - F) = -1$, which is equivalent to $S_i$ minus 0.5; if $S_i > F$, $Sgn(S_i - F) = 1$, which is equivalent to $S_i$ unchanged).

In theory, the output of the hidden layer will not exceed four (the output value of each hidden layer node is obtained according to the improved Sigmoid function. The value of the Sigmoid function does not exceed one, and the four hidden layer nodes mean that the input value of the output layer does not exceed four), and the output of the same category iris in the hidden layer should be larger than the output value of different categories iris (because the function is designed to make the same category of output values be closer to 1). Therefore, in order to make the output value of the same category larger, the input value of the output layer is the difference value between the sum of hidden layer outputs and four. Because the hidden layer outputs of the same category are larger, the gap with four is smaller, and the output value of output layer is closer to 1.

Z is brought into the Sigmoid function to get the output value $T$. Threshold $M$, $F$, and threshold weight $D_1$ and $D_2$ are optimized by the gradient inversion method [35]. The initial parameter values are summarized in the training results of the training iris (the irises in the same category that meet the prerequisites of this paper). After replacing the new training iris, according to the gap between the expected result and the actual result, the convergence is accelerated by the gradient descent, and then the parameter values are constantly adjusted. Compared with the optimization of existing frameworks

such as CNN, this paper uses the specially designed certification function of its own to perform threshold optimization on lightweight data with less calculation.

By designing various threshold parameters, the algorithm can adjust the input and output of each layer node in the neural network, so that the input between the layers of the same category of iris is as close as possible to negative infinity in the Sigmoid function and the output is as close as possible to 1. Finally, the purpose of expanding the difference among different categories of irises and narrowing the difference among the irises of the same category is achieved, thereby reducing the influence of iris deflection on the recognition effect.

### 3.3. Decision Layer

The image processing layer can extract the original stable map points for defocused iris conditions; the algorithms in the feature recognition layer are designed to eliminate the effects of iris deflection as much as possible. However, the accuracy of recognition is not only affected by the image itself, but also has a great correlation with device parameters, collector proficiency and external illumination environment. Therefore, the same algorithm may produce different recognition results in different external environments. In the case of ensuring the same collection equipment and only consider the effects of illumination, in the decision layer, two algorithms of feature recognition layer are tested and analyzed the experimental results to judge the credibility of the two algorithms' recognition results. The certification conclusion can be obtained through maximum credibility.

In the output of feature recognition layer, the results offer two outcomes: first, the test iris belongs to a specific category in the template iris library; second, the test iris does not belong to any category in the template iris library. Based on these two cases, the credibility of each recognition algorithm in the feature recognition layer is calculated separately under certain illumination conditions.

1. The test iris belongs to a specific category in the template iris library

Select $m$ test irises, denoted as $T1_i$. Each test iris corresponds to a template iris of the same category, consisting of $m$ template irises, denoted as $M1_i$. Comparing each test iris with all template irises, the sum of the cases in the recognition algorithm where the same category of iris is identified and different categories are excluded is calculated. The algorithm credibility is the ratio of the sum value and the total number of comparisons. The algorithm credibility $w_i$ is calculated by Equation (8).

$$w_i = \frac{\left(1 + \sum_{i=1}^{m} Sgn(F(T_i, M_i))\right)/2 + \left(1 + \sum_{y=1}^{m,y \neq i} \sum_{i=1}^{m} Sgn(F(T_i, M_y))\right)/2}{m \times m}, \tag{8}$$

where $F_1(T1_i, M1_i)$ is the result of the same category comparison.

2. The test iris does not belong to any category in the template iris library

Select $t$ test irises, denoted as $T2_i$. Different types of irises with test iris make up $r$ template iris, denoted as $M2_i$. Comparing each test iris with all template irises, the sum of cases where the different categories are excluded in the recognition algorithm is calculated. Then, Equation (9) is used to calculate the algorithm credibility $k_i$.

$$k_i = \frac{\left(1 + \sum_{y=1}^{r} \sum_{i=1}^{t} Sgn(F(T2_i, M2_y))/2\right)}{t \times r}, \tag{9}$$

where $F_2(T2_i, M2_y)$ is the algorithm comparison result between the test iris and the template iris.

When making the decision vote, the result with the most credibility is selected as the certification conclusion based on the credibility of each recognition algorithm under specific illumination conditions. The flexibility of the algorithm's influence on illumination and the correct recognition rate of the algorithm can be effectively improved by using this method.

## 4. Experiments and Analysis

**Data acquisition:** The JLU iris library of Jilin University, China [36] and CASIA iris library of Chinese Academy of Sciences, China [37] are used in all experiments in this paper. The JLU iris library is collected by the Biometrics and Information Security Technology Laboratory of Jilin University and generated by video screenshots. As of 2019, there are more than 80 categories of irises in the original iris library, and each category has more than 1000 images of various states. The number of images of unsteady irises is still expanding. The CASIA Iris Library is a commonly used iris library in the world of iris recognition and has been released for four generations.

**Experimental external environment:** In the experiments, the CPU frequency is dual-core 2.5 GHz, the memory is 8 GB, and the operating system is Windows.

**Evaluation metrics:** The evaluation indexes are the ROC curve [38] (a curve indicating the relationship between the False Reject Rate (FRR) and the False Accept Rate (FAR)), the Equal Error Rate (EER) (the value where FRR is equal to the FAR), and Correct Recognition Rate (CRR) [39].

**Experimental setup:** The experiments first self-certified the contribution of each layer to the certification and explained the relationship between the algorithm settings of each layer and the actual prerequisites. As set out in Section 4.1, the image processing layer experiment is mainly concerned with the extraction of defocused conditions, and explores the effect of extracted iris texture on iris certification; as set out in Section 4.2, feature recognition layer experiment was mainly designed to conduct tests on iris deflection and explore its impact on iris certification. As explained in Section 4.3, the decision layer experiment aimed mainly to test for the presence or absence of illumination and explore the impact of the statistical decision-making method based on external illumination on iris certification. Finally, a comprehensive experiment is set out in Section 4.4. As set out in Section 4.4.1, the actual effect of the overall model is tested, which can explain how the algorithm structure improves the overall recognition rate in the case where the traditional algorithms are used in each layer. As set out in Section 4.4.2, the performance of the algorithm is analyzed by comparing it to existing algorithms according to the prerequisites of this paper.

### 4.1. Image Processing Layer Experiment

The image processing layer experiment is mainly focused on the influence of iris texture on iris certification under defocused conditions. A recognition algorithm and decision layer were adopted in this paper's algorithms. The experimental iris library was the JLU-5.0 iris library.

The number of comparisons in the iris library is shown in Table 1.

**Table 1.** Number of comparisons of image processing layer experiment.

| Category Number | Iris Images per Category | Total Number of Images | Matches within Category | Matches outside Category | Total Number of Matches |
|---|---|---|---|---|---|
| 100 | 10 | 1000 | 1600 | 5130 | 6730 |

Compared the algorithm (ensemble model) in this paper's image processing layer with these following cases:

1. Do not process the iris, test iris recognition without noise suppression; (No processing)

2. Only use Gaussian filter to treat the iris, test iris recognition in the case of a single noise suppression algorithm; (Gaussian filtering)

3. The Log operator [32] and equalization histogram are used to process the iris, and the iris recognition under the combination of algorithms will be tested. (Log Operator and Equalization Histogram)

The CRR and EER in the four cases are shown in Table 2. The ROC curve of JLU-5.0 is shown in Figure 12.

In the image processing layer experiment, as can be seen from the results of Figure 12 and Table 2, the iris image processed in this paper has the highest recognition rate and the lowest EER after iris feature

extraction and recognition. This is because the image processing layer is designed for feature recognition services, which itself is used to preserve the feature points in the iris texture as much as possible, and then reflect the iris features according to the change rule of the amount of information of different regions.

**Table 2.** The CRR and EER in image processing layer experiment.

| No Processing | | Gaussian Filtering | | Log Operator and Equalization Histogram | | Ensemble Model | |
|---|---|---|---|---|---|---|---|
| CRR | EER | CRR | EER | CRR | EER | CRR | EER |
| 97.45% | 2.88% | 98.76% | 1.43% | 99.03% | 0.89% | 99.41% | 0.48% |

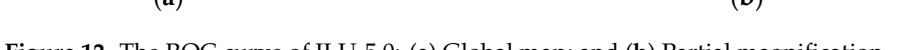

(**a**)        (**b**)

**Figure 12.** The ROC curve of JLU-5.0: (**a**) Global map; and (**b**) Partial magnification.

In the constrained iris, the blurred iris makes the definition between the texture and the non-texture unclear, and the effective feature information will be filtered, which can cause the change in the amount of information to be unrealistically reflected. Therefore, the filter algorithm of the image processing layer has a positive significance for the feature expression of the blurred image. The algorithm can filter out the non-textured regions, leaving the purest. A few image processing algorithms can also extract enough effective iris information to identify and improve the accuracy. However, too few processing algorithms are not representative, which is not conducive to find the feature points inside the iris; however, for many filter algorithms, the feature processing is too cumbersome, and the iris texture is also reduced. In this algorithm, six different angles of iris image processing algorithms are used to maximize the effective iris information and remove redundant parts to improve the recognition accuracy.

However, due to the algorithm's inability to fully control the external environment and the iris deflection problem, the effect of improvement may be limited.

*4.2. Feature Recognition Layer Experiment*

The feature recognition layer experiment was mainly used for testing iris deflection, and explored the effect of iris deflection on iris certification. Except for the algorithm in this paper, the other comparison algorithms directly use enhanced images for recognition, and do not provide a decision layer. The JLU-6.0 iris library of Jilin University was taken as the experimental iris library. The number of comparisons in the iris library is shown in Table 3.

**Table 3.** Number of comparisons in feature recognition layer experiment.

| Category Number | Iris Images in per Category | Total Number of Images | Matches within Category | Matches outside Category | Total Number of Matches |
|---|---|---|---|---|---|
| 50 | 100 | 5000 | 8765 | 20623 | 29388 |

The algorithm (Ensemble model) in this paper's feature recognition layer was compared with the following cases:

1. The iris recognition algorithm based on Zernike moment phase feature [40];
2. The iris recognition algorithm based on deep learning architecture [41];
3. The iris recognition algorithm based on statistically characteristic center symmetric local binary pattern (SCCS-LBP) [42];
4. The secondary iris recognition algorithm based on BP neural networks [11];
5. The iris recognition algorithm based on cross-spectral matching [43];
6. Iris recognition based on the Gabor + Hamming algorithm in the recognition layer of this study;
7. Iris recognition based on the Haar + BP algorithm in the recognition layer of this study.

The CRR and EER in the eight algorithms are shown in Table 4. The ROC curve of JLU-6.0 is shown in Figure 13.

**Table 4.** The CRR and EER in feature recognition layer experiment.

| Zernike | | Deep Learning Architecture | | SCCS-LBP | | Gabor + Hamming | |
|---|---|---|---|---|---|---|---|
| CRR | EER | CRR | EER | CRR | EER | CRR | EER |
| 98.73% | 1.47% | 96.78% | 3.58% | 97.43% | 2.63% | 97.79% | 2.31% |
| Secondary Iris Recognition | | Cross-Spectral Matching | | Ensemble Model | | Haar + BP | |
| CRR | EER | CRR | EER | CRR | EER | CRR | EER |
| 99.14% | 0.84% | 97.98% | 2.05% | 99.48% | 0.57% | 98.18% | 1.79% |

(**a**)            (**b**)

**Figure 13.** The ROC curve of JLU-6.0: (**a**) Global map; (**b**) Partial magnification.

In the feature recognition layer experiment, the results of Figure 14 and Table 4 are analyzed by the certification situation of the same person with the deflected iris: the certification accuracy of the certified structure is the highest, the EER is the lowest, and the ROC curve is closer to the horizontal and vertical coordinates axes. This is because, after filtering the texture information through the image processing layer, it is necessary to express the features of the iris according to the change rule of the amount of information of each part. Therefore, for the feature extraction of the constrained iris, the deflection state itself will have a certain impact on the feature extraction, because the deflection of iris will change the relative position of iris texture to some extent and affect iris features variation.

The traditional feature recognition algorithms mainly use statistical learning to carry out feature learning and recognizer training for ideal situations. On the one hand, this approach depends on the accumulation of several data and the accurate classification of data; on the other hand, it depends on the quality of the iris image itself. For example, for the deep-learning algorithm and secondary recognition neural network algorithm, in the case of insufficient number of iris and imperfect iris classification, it is difficult to improve recognizer structure by learning. Therefore, the learning of these algorithms can only be done through the existing ideal iris, the recognition rate is not as high as the algorithm in this paper. In the feature recognition layer experiment. Case 1, Case 3, and Case 5 have higher requirements for the quality of the iris itself. In the case of defocusing and deflection, the recognition accuracy is lower than this study's algorithm. In this paper, the recognition algorithms are selected based on the results of comprehensive analysis based on the current environment, so the recognition algorithms are not fixed. Therefore, the final judgment result of the comprehensive feature recognition layer is better, and the applicable iris range is wider.

### 4.3. Decision Layer Experiment

The decision layer experiment was mainly focused on the experiment of illumination and explored the impact of decision methods on iris certification based on test statistics according to external lighting conditions. The comparison algorithms directly used the enhanced iris image for identification. The JLU-3.5 iris library of Jilin University was taken as the experimental iris library.

The algorithm (Ensemble model) in this study's feature recognition layer was compared with the following cases:

1. The feature recognition layer is composed based on the comparative experiment 1–5 in Section 4.2. The decision is made by most decision voting methods [44]; (Posterior probability decision)

2. The feature recognition layer is composed based on the comparative experiment 1–5 in Section 4.2. The weight ratio is set for each recognition algorithm by test experiment, and the method of cumulative majority vote is used [45]; (Cumulative sums and majority vote)

3. Multimodal ocular biometric in visible spectrum based on posterior probability method [46]. (Posterior probability decision)

The number of comparisons in the iris library is shown in Table 5. The CRR and EER are shown in Table 6. The ROC curve of JLU-3.5 is shown in Figure 14.

**Table 5.** Number of comparisons in decision layer experiment.

| Category Number | Iris Images in per Category | Total Number of Images | Matches within Category | Matches outside Category | Total Number of Matches |
|---|---|---|---|---|---|
| 100 | 20 | 2000 | 2645 | 8156 | 10801 |

**Table 6.** The CRR and EER in decision layer experiment.

| Decision Voting | | Posterior Probability Decision | | Cumulative Sums and Majority Vote | | Ensemble Model | |
|---|---|---|---|---|---|---|---|
| CRR | EER | CRR | EER | CRR | EER | CRR | EER |
| 98.42% | 1.81% | 97.83% | 2.64% | 95.76% | 4.54% | 99.18% | 0.72% |

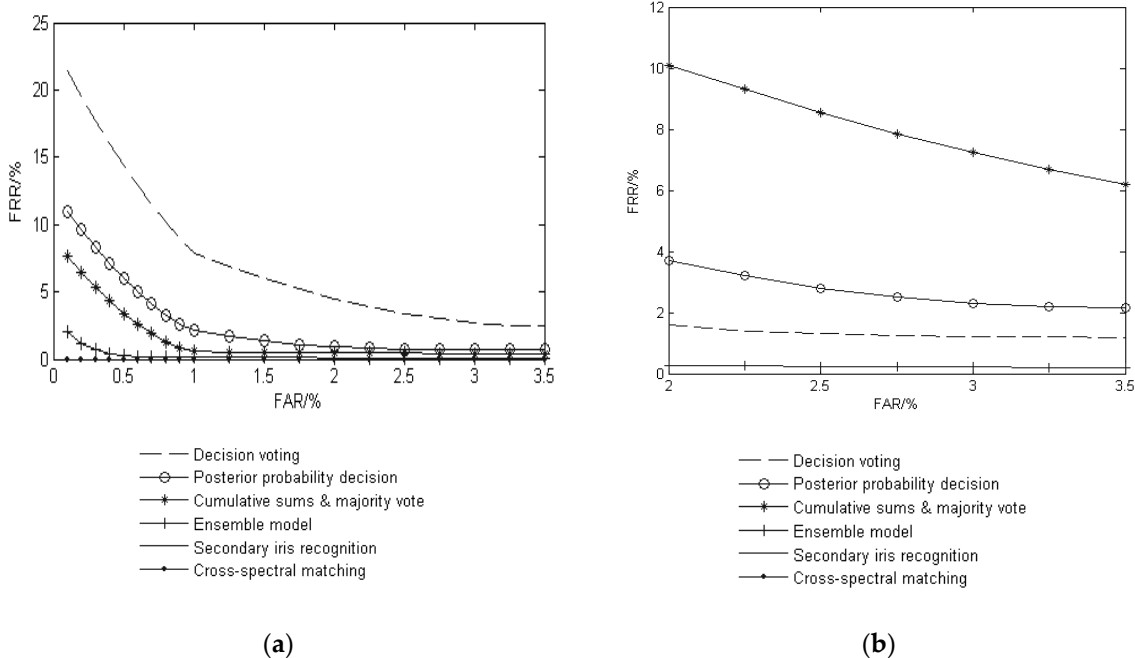

**Figure 14.** The ROC curve of JLU-3.5: (**a**) Global map; and (**b**) Partial magnification.

In the decision layer experiment, as can be seen from the results of Figure 14 and Table 6: the certification structure of this paper has the highest recognition rate, the lowest EER, and the ROC curve is closer to the horizontal and vertical axes. Although the image processing layer and the feature recognition layer can effectively eliminate the blurred image and the influence of the deflection, the illumination effect is difficult to exclude, because the fixed threshold cannot accurately quantify the influence of the illumination, so the intensity of illumination can affect the appearance of iris features to a certain extent, and can also affect the amount of information in a certain area. The recognition accuracy of the decision-level fusion strategy algorithm, which causes the posterior probability iris, sclera and periocular features is also be affected. Most voting methods are applicable to the case where there are many recognition algorithms, which will slow down the speed of certification. At the same time, through the feature recognition layer experiment, it can be seen that single recognition algorithms also received interference from the external environment, which may be a leak in the recognition of voting. The cumulative voting method based on the weight ratio is dependent on the setting of the feature ratio. Both cumulative voting and majority voting can potentially take the occasional case as statistical data, which may lead to identification error.

In this paper, the decision layer adopted a method of trusted analysis. The statistical algorithm is recognized under different illumination conditions, and then it is analyzed which algorithm is more reliable in one certain specific illumination condition. Although the decision conclusion is also obtained through statistics, in the process of analysis, the interference of accidental conditions can be excluded and the decision for iris recognition conclusions can also be made in a more credible way with respect to the environment. The algorithm has more environment inclusiveness and the result is more convincing.

### 4.4. Certification Structure of Comprehensive Effect Experiment

The experiment is divided into two parts. In Section 4.4.1, the effect of the algorithm on the overall performance improvement can be explained by determining the meaning of tightness degree among each layer. In Section 4.4.2, it is compared with other certification algorithms to find out why, in the illustrated case of using a traditional algorithm to fill the multi-algorithm parallel structure, this algorithm is able to achieve a high recognition rate under the prerequisites of this paper.

The irises that meet the prerequisites are selected from the CASIA-V1, CASIA-Iris-Interval, and CASIA-Iris-Lamp iris libraries, which are taken as experimental iris library. The overall performance of this algorithm is judged by experimental results. The number of comparisons of the iris library in the certification structure comprehensive effect experiment is shown in Table 7.

**Table 7.** Number of comparisons in comprehensive experiment.

| Iris Library | Category Number | Iris Images in per Category | Total Number of Images | Matches within Category | Matches Outside Category | Total Number of Matches |
|---|---|---|---|---|---|---|
| CASIA-V1 | 120 | 5 | 600 | 1200 | 6000 | 7200 |
| Iris-Interval | 200 | 5 | 1000 | 2000 | 5460 | 7460 |
| Iris-Lamp | 400 | 5 | 2000 | 3500 | 6500 | 10000 |

### 4.4.1. Meaning of Tightness Degree for Each Layer

The CRR and EER of the meaning of tightness degree are shown in Table 8.

**Table 8.** The CRR and EER of the meaning of tightness degree.

| CASIA-V1 | | CASIA-Iris-Interval | | CASIA-Iris-Lamp | |
|---|---|---|---|---|---|
| CRR | EER | CRR | EER | CRR | EER |
| 99.78% | 0.38% | 99.37% | 0.52% | 99.13% | 0.85% |

It can be seen from Tables 7 and 8 that the correct recognition rate of the algorithm can reach more than 99%, and the EER is also maintained at a low level. Although this paper is used to fill the multi-algorithm parallel structure using traditional existing algorithms, it can still make the final certification results achieve high results according to the prerequisites set out in this paper.

Because the iris recognition itself is a holistic model, the connection of each step is interactive. Based on the premise of ensuring that the processing algorithm has no negative impact on iris certification, the combination connection between the feature extraction and recognition algorithms is extremely important.

It can be seen in the experiments described in Section 4.1, Section 4.2, and Section 4.3 that after three layers are integrated, the recognition accuracy can reach a higher degree, the structure of each layer can effectively suppress the influence of negative factors. Because this paper considers the problem of algorithm connection in the three-layer structure design, in order to accurately extract the points in the iris texture, six algorithms are used for image processing. On the one hand, it avoids insufficient image noise suppression caused by too few image processing algorithms; on the other hand, it avoids excessive image processing algorithms and leads to an insufficient amount of effective information; the Gabor filter and the Haar wavelet are used to express the change rule of information in each region, and the information amount is used as the iris feature. The certification result is obtained by matching the corresponding certification algorithm. Although the recognition algorithm itself needs to be optimized according to the features, the data itself is not complicated and the optimization parameters are less, so compared with the complex structure such as CNN, the traditional gradient method can get good training results. Finally, the result trust degree of each algorithm is obtained under different illuminations, and the algorithm result that get maximum trust is used as the final certification result. The three layers are closely connected and interact with each other, achieving the improvement of the comprehensive performance of the simple algorithm.

4.4.2. Existing Method Comparison Experiment

The iris used in the comparative experiment is the iris recognition area (256 × 32 dimension), and the iris recognition is shown in Table 7 (the same as in Section 4.4.1). The number of training irises is 20% of the number of test irises, and the training irises are not the same as the test irises.

1. The certification algorithm is compared with the following seven algorithms.

2. Concept cognition based on deep learning neural network [47], compared with deep learning architecture cognitive model for a small number of samples;

3. Evidence theory certification by clustering method [48], compared with certification model under clustering feature;

4. Iris certification algorithm of traditional convolutional neural network [49], compared with the mode of traditional convolutional neural networks trained;

5. A certification function optimization algorithm based on decision particle swarm optimization algorithm and stable feature [50], compared with certification model with improved certification function;

6. Iris recognition algorithm based on multi-feature weighted fusion [51], compared with traditional multi-source feature algorithm;

7. Statistical learning model of convolutional neural networks based on VGG16 model architecture [52];

8. Faster R-CNN Inception Resnet V2 model architecture [53] for iris certification.

The CRR of the three iris libraries compared to the existing algorithms are shown in Table 9.

**Table 9.** The CRR of the three iris libraries compared to the existing algorithms.

| Iris library | CASIA-V1 | CASIA-Iris-Interval | CASIA-Iris-Lamp |
|---|---|---|---|
| Deep learning neural network [47] | 76.68% | 74.69% | 79.84% |
| Evidence theory certification by clustering method [48] | 86.45% | 83.78% | 81.76% |
| Traditional convolutional neural network [49] | 83.12% | 85.46% | 78.42% |
| Decision particle swarm optimization algorithm and stable feature [50] | 91.23% | 90.75% | 88.73% |
| Multi-feature weighted fusion [51] | 69.75% | 70.86% | 73.42% |
| VGG16 [52] | 80.74% | 83.47% | 84.23% |
| Faster R-CNN Inception Resnet V2 [53] | 93.42% | 87.65% | 94.31% |
| Algorithm in this paper | 99.78% | 99.37% | 99.13% |

As can be seen from Table 9, the algorithm has a strong advantage in the lightweight iris certification process. The deep learning algorithm requires a large amount of data and a sufficiently clear classification label. The amount of data used for learning in this paper is obviously insufficient, so that the final certification effect does not exert the ability of deep learning, which leads to the low certification accuracy. The clustering method can distinguish the differences between the irises by clustering the same category of iris features, which can be proposed for an optimal clustering effect in the current situation. However, in the case of interference and low numbers of training irises, for part of the iris, the feature clustering effect is not obvious, which also makes the algorithm's certification accuracy lower.

The decision particle swarm optimization algorithm and the stability feature algorithm are designed for the prerequisites of this paper. By modifying the certification function and improving the environmental inclusiveness of the model, the recognition accuracy is improved. However, this algorithm requires a large amount of iris data for training; for lightweight iris data, the algorithm is insufficiently trained for different iris states, so the recognition accuracy is not as good as the algorithm in this paper. The multi-feature weighted fusion recognition algorithm uses the Haar and LBP algorithm to extract iris features, and the Hamming distance is used for identification; the weight relationship between the two methods is set according to the statistical results of training data. This

algorithm is a multi-source feature extraction for the ideal state iris library. However, compared with the multi-state iris certification, the Hamming distance is less inclusive for the environment and is not suitable for the iris that has a large change in the iris state. Therefore, the recognition accuracy is lower in the multi-state iris.

As a common deep learning architecture, convolutional neural network is commonly used in various biometric recognitions, such as face recognition [54], palmprint recognition [55], fingerprint recognition [56], and iris recognition [57]. The feature cognition of the traditional convolutional neural network is to express the iris category by setting the label. This type of method makes the iris representative, but it is affected by unsteady iris state. Such labels are very unstable under lightweight data, so the recognition accuracy is unstable. Both the VGG16 model and the Inception model are currently the most commonly used convolutional neural network models, which show very good performance in feature certification; however, this does not mean that they are suitable for the iris data in this paper.

The VGG16 model is composed of 13 convolutional layers and 3 fully connected layers. In the iris recognition, the iris image is continuously convolved, pooled, and ReLU processed to obtain different types of feature values, thereby achieving iris recognition. However, due to the prerequisites in this paper, the state of iris itself is unpredictable. It can be seen from the different iris states in Figure 1. In cases where iris feature extraction can be performed, without the detailed classification of iris training data and a large amount of iris data, this unpredictable state makes it difficult to take advantage of the superiority of the VGG16 model. In addition, the amount of training data in this paper is not much. The calculation of VGG16 model is redundant, and the excessive parameters increase the difficulty of training. These greatly reduce the certification accuracy of VGG16 model.

The Faster R-CNN Inception Resnet V2 model architecture is a more advanced convolutional neural network model that allows training on lightweight data compared to the VGG16 model, reducing the difficulty of training. However, under the prerequisites of this paper, this paper first adopts a variety of image processing algorithms to improve the connection between the image and the feature extraction algorithm, so that the conventional algorithm can achieve high recognition effect in the feature image. Although the R-CNN model reduces the complexity of training, it also counts the redundant features in the training. Compared with the stable features of this paper, the interference is formed. Considering the changes in the state of the iris acquisition and the conditions of the existing architecture, configure the corresponding image processing algorithm or directly input the stable features of the iris into the existing architecture may result in incompatibility between the image data and the architecture. These all reduce the certification accuracy of the model.

Therefore, it can be seen that the algorithm uses the connection between the layers to effectively link the image processing, feature extraction and iris certification. The problem of incompatibility between existing architecture requirement data and image processing can be avoided, and the potential of traditional algorithms can also be fully utilized. At the same time, it satisfies the training of lightweight data in a complex and variable acquisition state.

## 5. Conclusions

In this paper, aiming at the one-to-one certification problem of small-scale lightweight constrained irises with defocus, deflection, and illumination, based on existing algorithms and a multi-algorithm parallel integration model structure, a one-to-one certification algorithm based on multi-algorithm integration and maximum trusted decision was proposed. The image processing layer was used to suppress defocus interference, and a stable iris that satisfies the amount of effective information could be extracted. The feature recognition layer was used to suppress deflection interference, and two recognition results could be obtained. The decision layer suppressed the influence of illumination, the maximum credibility of the recognition algorithm was obtained by environmental test statistics, and the most credible result was used as the final decision result. In the small-scale and multi-state iris, the

efficiency of the existing iris algorithm was maximized, the training difficulty was reduced, and the certification accuracy was improved.

The algorithm structure connection in this paper also needed some improvements and the iris algorithm prerequisites involved in this paper were quite broad; however, this method had certain defects for large-scale iris certification and multi-category classification, making it only applicable to lightweight certification. Investigating how to further improve the recognition accuracy based on the prerequisites for expanding the recognition algorithm and achieving multi-category classification will be the focus of future research.

**Author Contributions:** Conceptualization, L.S. and Z.X.; Data curation, L.S.; Formal analysis, L.S., L.Y. and Z.X.; Funding acquisition, L.Y. and Z.X.; Investigation, L.S.; Methodology, L.S.; Project administration, L.S. and D.T.; Resources, L.S., Z.K., D.T., L.X. and W.C.; Software, L.S., Z.K. and W.C.; Supervision, L.S.; Validation, L.S.; Visualization, L.S.; Writing—original draft, L.S.; Writing—review & editing, L.S.

**Funding:** This research was funded by the National Natural Science Foundation of China (NSFC), grant number 61471181; Natural Science Foundation of Jilin Province, grant number 20140101194JC and 20150101056JC; Jilin Province Industrial Innovation Special Fund Project, grant number 2019C053-2. Thanks to the Jilin Provincial Key Laboratory of Biometrics New Technology for supporting this project.

**Conflicts of Interest:** The authors declare no conflict of interest.

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
