# Peer review of "Unsteady State Lightweight Iris Certification Based on Multi-Algorithm Parallel Integration"

_algorithms, doi:10.3390/a12090194_

Round 1

Reviewer 1 Report

This paper proposed a multi-algorithm Integration and Maximum Trusted Decision for Unsteady State Iris Certification. Some issues need to be addressed more carefully.

The abstract in current form is kind of superficial, and the abstract needs to rewrite to point out the significance and impact of the paper. The introduction, the motivation of the paper needs to be articulated far more clearly. Moreover, in the introduction section, a more rigorous investigation on the existing methods, such as comparison of previous approaches in terms of pros and cons, should be given.  I found many English errors even in the abstract, so please go through the entire manuscript to double-check accuracy/consistency of each equation, table, figure and reference, and ensure English grammar error-free.   Line 228-229, why "2" is the starting letter of this the sentence?. I found many similar errors, please double check all the sentence after the captions (e.g. 362-363, 370-371) Line 410-411, the authors must explain more about the experimental results in Figure 14 and Table 2. Similar case for Table 3, Figure 15, Figure 16, Table 7, Table 8, and Figure 19. The authors cannot expect the readers to interpret these results by themselves. In my opinion, there are too many figures in this research. The authors should reconsider combining or removing them, such as figure 4, figure 5, figure 9, figure 10, figure 11. There should be more explanation for the two datasets used in this research. A separate section for dataset would be better. Please explain carefully how Gabor and Hamming's parameters affect the performance of the system (sensitivity) Iris recognition topic belongs to biometric identification methods. So I recommend the authors add some more recent studies related to forensic face in this paper

Tome, Pedro, Ruben Vera-Rodriguez, Julian Fierrez, and Javier Ortega-Garcia. "Facial soft biometric features for forensic face recognition." Forensic science international 257 (2015): 271-284.

Dang, L., Syed Hassan, Suhyeon Im, Jaecheol Lee, Sujin Lee, and Hyeonjoon Moon. "Deep learning based computer generated face identification using convolutional neural network." Applied Sciences 8, no. 12 (2018): 2610. 

Author Response

The attachment contains three parts

1, a point-by-point response to your comments

2, the modification of your question in the article 

3, MDPI modified English certificate

Reviewer 2 Report

In the paper, a method for iris identification is presented. The approach jointly uses several algorithms to obtain a representation which exhibits a promising accuracy. However, due to the lack of clarity, superficial discussions, and problems with English writing, the paper should be completely rewritten and completed. This would allow determining whether it contributes to the state-of-the-art.
The detailed comments are as follows:
1.     The linguistic quality of the paper makes its findings difficult to follow.  The sentences are lengthy, repetitive, confusing, and contain a lot of mistakes and typos. For example, Abstract contains only five sentences.
2.     Please add at least 10 recent works on iris identification (apart from Authors’ papers).
3.     The contribution of the paper is not clear. Which parts are novel? Which parts overlap with previous Authors’ works (e.g., [27] and [18])? It seems that previous papers share some common findings.
4.     Abstract. What ‘satisfying the effective information amount’ means? Such criteria should be formally defined. What information is regarded as effective?
5.     Abstract. The “maximum credibility of recognition algorithms” is not defined in the paper. Here, an optimization problem should be formulated. It the solution optimal?
6.     Please explain the difference between ‘certification’ and ‘authentication’.
7.     Since the combination of different algorithms is used, their individual contributions to the performance should be analyzed.
8.     It seems that Authors used many image processing algorithms to obtain features and the rest is handled by the neural network. Such an approach is not novel. However, it is interesting, since the results are quite promising. There are many other image processing/filtering operators. Why only six of them were used?
9.     The feature stability requires discussion. Instead of showing the readers figures with iris textures, it would be advisable to show the features and discuss the range of their change. It does not show how affected they are by illumination or blur.
10.  The method uses many parameters. Their impact on performance is not discussed.
11.  Each considered case should be clearly addressed (defocus, deflection, etc.). The structure of the experimental part of the paper requires substantial changes, explaining what would be tested and why.
12.  The results in tables and in Figures are not commented (pages 12-16).  The analysis in Section 4.5 is superficial. It would be better to analyze the results where they are shown. Here, a further discussion, as well as limitations of the method should be elaborated.
13.  The method is not compared with the state-of-the-art algorithms. What are its advantages? Why readers should apply the presented findings into their methods?
14.  Conclusions state that the paper requires structure modifications. Please make them.

Author Response

(The authors gave the same response as above.)

Reviewer 3 Report

This paper proposes a multi-algorithm framework for iris recognition, when different images are captured at different conditions. They analyze different choices of image processing step, as well as feature extraction, and also predictor step. There some concerns with this work which need to be addressed:

This work analyzes several classical approaches for pre-processing, as well as feature extraction, and decision aggregation on iris recognition. However most of these techniques pretty well-known and have been investigated before. Ideally more technical contribution is expected from a journal paper. I would suggest the authors to somehow distinguish their work from previous ones. Also, it is not clear what is the criteria for choosing different algorithms at each step? For example why specific pre-processing filters are evaluated, and some of the other algorithms are not chosen? Are these based on some previous validation? How are the parameters of this framework tuned? Is that based on some tuning on a separate validation set? Also the paper title should be re-worded a little bit to be more fluent and readable. Instead of iris shooting, I suggest to use “iris image acquisition”, which is more technical alternative. In the experimental result section, this work skips comparing with many of the recent state-of-the-art iris recognition algorithms. Many of the recent works on iris recognition and biometric recognition are missing from introduction, and references. I suggest authors to do a more thorough literature search and add more relevant works. Some of them are mentioned below:

[a] "Iris biometrics using deep convolutional networks." 2018 IEEE International Instrumentation and Measurement Technology Conference (I2MTC). IEEE, 2018.

[b] "An experimental study of deep convolutional features for iris recognition." 2016 IEEE signal processing in medicine and biology symposium (SPMB). IEEE, 2016.

[c] "An end to end deep neural network for iris segmentation in unconstrained scenarios." Neural Networks 106 (2018): 79-95.

[d] "Iris recognition using scattering transform and textural features." 2015 IEEE signal processing and signal processing education workshop (SP/SPE). IEEE, 2015.

Author Response

This work analyzes several classical approaches for pre-processing, as well as feature extraction, and decision aggregation on iris recognition. However most of these techniques pretty well-known and have been investigated before. Ideally more technical contribution is expected from a journal paper. I would suggest the authors to somehow distinguish their work from previous ones.

Author response:

This paper modifies the Section”introduction ” and clarifies the following points.

What method is proposed for this purpose?(line 90,page 3) What is the key to this method and what is the point?(line 95-123,page 3)

3, this paper contributes and innovates(line 124-139,page 4)

Modified the structure of the article, istinguish our work from previous ones.

Also, it is not clear what is the criteria for choosing different algorithms at each step? For example why specific pre-processing filters are evaluated, and some of the other algorithms are not chosen?Are these based on some previous validation? How are the parameters of this framework tuned? Is that based on some tuning on a separate validation set?

Author response:

For the reason of algorithm selection, the Section 3 of this paper has added a supplementary explanation(Page 7-8 ;line224-269). The algorithms selected in this paper are fully certified algorithms that are useful for iris recognition. The main consideration is the connection between the algorithms and the degree of connection between iris feature extraction and certification.

Also the paper title should be re-worded a little bit to be more fluent and readable. Instead of iris shooting, I suggest to use “iris image acquisition”, which is more technical alternative. In the experimental result section, this work skips comparing with many of the recent state-of-the-art iris recognition algorithms. Many of the recent works on iris recognition and biometric recognition are missing from introduction, and references. I suggest authors to do a more thorough literature search and add more relevant works.

Author response:

Added experiment with the existing algorithm, (line666), the title is also be modified, added the attribute "lightweight" to better reflect the prerequisites of this article.(literature 46-49)

In the experimental result section, this work skips comparing with many of the recent state-of-the-art iris recognition algorithms. Many of the recent works on iris recognition and biometric recognition are missing from introduction, and references. I suggest authors to do a more thorough literature search and add more relevant works.

Author response:

Literatures 3,4,7,8,9,10,12,25,28,37,45,46,47 are new literatures, a total of 13 recent literatures

The Literatures a, b, and c mentioned by the author have been added (literatures 48, 46, 4).

This paper has greatly revised the Section "Experiment and Analysis"  

As set out in Section 4.1, the image processing layer experiment is mainly concerned with the extraction of defocused conditions, and explore the effect of the extracted iris texture on iris certification. As set out in Section 4.2, feature recognition layer experiment is mainly designed to conduct tests on iris deflection and explore its impact on iris certification. As explained in Section 4.3, the decision layer experiment aimed mainly to test for the presence or absence of illumination and explore the impact of the statistical decision-making method based on external illumination on iris certification. Finally, in the comprehensive experiment set out in Section 4.4, the actual effect of the overall model is tested, which could explain how the algorithm structure improved the overall recognition rate in cases where traditional algorithms are used in each layer. In addition, the performance of the algorithm is analyzed by comparing it with existing algorithms according to the prerequisites of this paper.

Reviewer 4 Report

In this paper, the authors presented multi-algorithm parallel integration model for iris recognition problem. The idea is interesting, but the article has many problems and its structure should be improved.

Minor:
1) Adding images in the introduction section is unnecessary.
2) Captions, for example for images, do not have punctuation and spaces.
3) Authors should pay more attention to machine learning in the introduction:
a)
b)
4) Each section should begin with a text, not a subsection.

Major:
1) There is no mathematical model. Graphic processing is omitted completely in terms of description.
2) Figure 2 should be better visualized, maybe as a sequence of actions?
3) The mathematical model of Gabor and Haar as well as other elements are missing. The description is not what is described in section 3.3.1. Authors should correctly enter the designs so that the work is smooth and legible.
4) Mathematical formulas should be better formatted.
5) The authors take many factors automatically, as seen in line 266. Where is the analysis of the selection of these values? Such elements should be in the experimental section. In the current version, the authors do not adhere to the classic article division, which should be repaired.
6) How is the network taught? How was the architecture chosen? Have other been tested?
7) Why is this solution better than CNN, or faster RCNN? There are no adequate comparisons.

Author Response

Minor:

1) Adding images in the introduction section is unnecessary.

2) Captions, for example for images, do not have punctuation and spaces.

3) Authors should pay more attention to machine learning in the introduction:

a)

b)

4) Each section should begin with a text, not a subsection.

Author response:

Introduction chapter image has been adjusted, modified the article structure The image has been modified to remove inappropriate punctuation and spaces. The introduction has been revised to highlight the structural purpose and experimental results of this paper.

4.Each part starts with text instead of the beginning of the bar

Major:

There is no mathematical model. Graphic processing is omitted completely in terms of description.

Author response:

The introduction of the algorithm has been modified, especially in terms of Gabor and Haar wavelets, and a natural description of the mathematical language has been carried out. The significance of the parameters in Gabor filtering and Haar and the influence of parameters on the experiment are better highlighted.(highlight,Page9-12,line317-337;342;398-403;419-423)

Figure 2 should be better visualized, maybe as a sequence of actions?

Author response:

Figure 2 (Modified as Figure 1)shows the phased results of the various parts of the whole pretreatment process. This paper re-introduces the pretreatment process and modifies the text of the diagram.(Page 4,line 141-149)()

The mathematical model of Gabor and Haar as well as other elements are missing. The description is not what is described in section 3.3.1. Authors should correctly enter the designs so that the work is smooth and legible.

Author response:

The revised article has greatly revised the description of Gabor and Haar, clarifying the meaning of the parameters, standardizing the actual process, and making the work clearer.

Mathematical formulas should be better formatted.

Author response:

Mathematical formulas have been trimmed to better format

The authors take many factors automatically, as seen in line 266. Where is the analysis of the selection of these values? Such elements should be in the experimental section. In the current version, the authors do not adhere to the classic article division, which should be repaired.

Author response:

This paper add the process of parameter settings in the feature recognition layer in Section 3.(Page 10,line 343;Page 12,line 398; Page 12,line 418)

How is the network taught? How was the architecture chosen? Have other been tested?

Author response:

Network optimization is not the focus of this article.Added explanation about network optimization(Page 12,line 418)

Why is this solution better than CNN, or faster RCNN? There are no adequate comparisons.

Author response:

Added experiments compared with the prior art, illustrating the situation after comparison with traditional CNN(Page 20,line 666)

Round 2

Reviewer 1 Report

Thank you the authors for addressing all of my previous comments. There are some minor changes required for publication.

Figure 2 is not clear, please revise. Iris recognition topic belongs to biometric identification methods. So I recommend the authors add some more recent studies related to forensic face in this paper

Tome, Pedro, Ruben Vera-Rodriguez, Julian Fierrez, and Javier Ortega-Garcia. "Facial soft biometric features for forensic face recognition." Forensic science international 257 (2015): 271-284.

Dang, L., Syed Hassan, Suhyeon Im, Jaecheol Lee, Sujin Lee, and Hyeonjoon Moon. "Deep learning based computer generated face identification using convolutional neural network." Applied Sciences 8, no. 12 (2018): 2610. 

Author Response

The attachment contains two parts

1, a point-by-point response to your comments

2, the modification of your question in the article (different color labels)

Reviewer 2 Report

The revised manuscript is much clearer and better describes the findings. However, some minor issues should be taken into account:
1.      It is written that the method speeds up the certification. However, the speedup is not shown in the paper. Also, the speed of operation is not discussed. What does it mean that it is guaranteed?  Please provide a table with timings and saved time.  
2.      The word ‘stability’ means that the measured or obtained information is robust to some distortions. Hence, the stability of the features (page 8, line 271) must be examined. Also, the efficiency of the information must be clearly defined and shown.
3.      Authors are encouraged to provide a sourcecode of their method ensuring the repeatability of the presented results

Author Response

(The authors gave the same response as above.)

Reviewer 3 Report

Thanks to the author for revising this work.

I do not see many of my comments/concerns addressed in the revised version. Ideally a point-to-point response letter is expected from the authors when submitting the revised version, not just the highlighted manuscript.

I am going to mention my concerns again.

This work analyzes several classical approaches for pre-processing, as well as feature extraction, and decision aggregation on iris recognition. However most of these techniques pretty well-known and have been investigated before. Ideally more technical contribution is expected from a journal paper. I would suggest the authors to somehow distinguish their work from previous ones. Also, it is not clear what is the criteria for choosing different algorithms at each step? For example why specific pre-processing filters are evaluated, and some of the other algorithms are not chosen? Are these based on some previous validation? How are the parameters of this framework tuned? Is that based on some tuning on a separate validation set? Also the paper title should be re-worded a little bit to be more fluent and readable. Instead of iris shooting, I suggest to use “iris image acquisition”, which is more technical alternative. In the experimental result section, this work skips comparing with many of the recent state-of-the-art iris recognition algorithms. Many of the recent works on iris recognition based on deep learning are missing from introduction, and references. I suggest authors to do a more thorough literature search and add more relevant works. Some of them are mentioned below:

[a] "Iris recognition using scattering transform and textural features." 2015 IEEE signal processing and signal processing education workshop (SP/SPE). IEEE, 2015.

[b] "Convolutional neural network-based feature extraction for iris recognition." Int. J. Comp. Sci. Info. Tech. 10 (2018): 65-78.

[c] "DeepIris: Iris Recognition Using A Deep Learning Approach." arXiv preprint arXiv:1907.09380 (2019).

Author Response

(The authors gave the same response as above.)

Reviewer 4 Report

1) The abstract is too long and general.
2) The mathematical model should be written in more details. Especially, the training techniques.
3) The content of the parenthesis in equations should be included in relation to the height of the parenthesis itself.
4) Much more accurate comparative tests with CNN should be added, in particular analysis of tested architectures, time and accuracy measurement. Please include VGG16 and Inception.
5) The summary should emphasize the impact of work on CNN or RCNN.

Author Response

(The authors gave the same response as above.)

Round 3

Reviewer 1 Report

Thank you the authors for answering all of my questions. All of my comments have been addressed, so I recommend this study for publication

Reviewer 2 Report

Authors have addressed most of my comments.

Reviewer 3 Report

Thanks to the authors for addressing some of my comments. The paper is in a better shape now, compared to the first submitted version.

I think it can still be improved in terms of experimental results by adding more details, and comparison with previous works, and also some of the very recent works on iris recognition. But I leave that for authors to decide, as I think the current version could be published.

Reviewer 4 Report

It can be accepted in current form.